# Botnet Defense System: Concept, Design, and Basic Strategy †

**Shingo Yamaguchi** 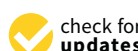

Graduate School of Sciences and Technology for Innovation, Yamaguchi University, Ube 755-8611, Japan; shingo@yamaguchi-u.ac.jp

† This paper is an extended version of our paper published in Yamaguchi, S. Botnet Defense System: Concept and Basic Strategy. In Proceedings of the IEEE 38th International Conference on Consumer Electronics (IEEE ICCE 2020), Las Vegas, NV, USA, 5 January 2020.

**Abstract:** This paper proposes a new kind of cyber-security system, named Botnet Defense System (BDS), which defends an Internet of Things (IoT) system against malicious botnets. The concept of BDS is "Fight fire with fire". The distinguishing feature is that it uses white-hat botnets to fight malicious botnets. A BDS consists of four components: Monitor, Strategy Planner, Launcher, and Command and Control (C&C) server. The Monitor component watches over a target IoT system. If the component detects a malicious botnet, the Strategy Planner component makes a strategy against the botnet. Based on the planned strategy, the Launcher component sends white-hat worms into the IoT system and constructs a white-hat botnet. The C&C server component commands and controls the white-hat botnet to exterminate the malicious botnet. Strategy studies are essential to produce intended results. We proposed three basic strategies to launch white-hat worms: *All-Out*, *Few-Elite*, and *Environment-Adaptive*. We evaluated BDS and the proposed strategies through the simulation of agent-oriented Petri net model representing the battle between Mirai botnets and the white-hat botnets. This result shows that the Environment-Adaptive strategy is the best and reduced the number of needed white-hat worms to 38.5% almost without changing the extermination rate for Mirai bots.

**Keywords:** IoT; cyber-security; botnet; malware; multi-agent system; Petri net

---

## 1. Introduction

The Internet of Things (IoT) is a fundamental technology which brings about radical change in modern society, while being targeted as a springboard for cyber-attacks. A new kind of malware called *Mirai* [1] infects IoT devices and turns them into bots. The network of bots (botnet) becomes a springboard for Distributed Denial-of-Service (DDoS) attacks. In fact, Mirai botnet's DDoS attacks came true in September 2016 and knocked out Amazon, Twitter, and other major sites. IoT devices are explosively increasing and most of them are vulnerable. Since Mirai makes such IoT devices a hotbed, Mirai botnet's DDoS attacks tend to become massive and disruptive [2]. Four years have passed since Mirai appeared, but Mirai and its variants are still raging all over the world [3].

Mirai penetrates only to the dynamic memory of a device, therefore we can delete Mirai by rebooting the infected device [4]. However, Moffitt [5] reported that Mirai can reinfect the device within minutes unless the vulnerability is patched. IoT devices are explosively increasing. According to the Cisco's white book [6], more devices were connected to the Internet than people by the late 2000s and the number is predicted to reach 30 billion by 2023. Also, IoT devices are vulnerable. This is because they do not have resources to run security functions and their vendors may sacrifice security in the price competition and/or their rush to market. We must fix such vulnerabilities, but workforce tactics

are not practical because of the huge number of devices. Thus, it is necessary to drastically increase the ability to defend IoT systems against Mirai.

Yamaguchi [7] has proposed to use worms to defend IoT systems by imitating the way that attackers use malware to attack. He defined a white-hat worm as a worm which drives out malicious botnets and then deletes itself. The worm is characterized by two attributes: *secondary infection possibility* and *lifespan*. Secondary infection enables the worm to regain the device from Mirai. Lifespan forces the worm to destruct itself and avoid staying on the recovered device. He regarded the battle between Mirai and the white-hat worm as a multi-agent system and expressed it with agent-oriented Petri nets, called $PN^2$ [8]. The simulation result of the $PN^2$ model showed the effectiveness of the worm. However, there is no discussion on how to systematically launch the white-hat worm in response to Mirai's infection situation.

In this paper, we propose a new kind of cyber-security system, named *Botnet Defense System* (*BDS*). In imitation of "Fight fire with fire", we advocate a concept of "Fight botnet with botnet". BDS is a system that realizes this concept. Next we describe the system organization and operations of BDS. BDS strategically operates the white-hat worm and its botnet in response to Mirai's infection situation. Also, we propose a basic strategy for launching white-hat worms. Then we evaluate the effect of the strategy through the simulation with the $PN^2$ model.

The rest of this paper is organized as follows: Section 2 surveys the related work. Section 3 gives the design of BDS and basic strategies. Section 4 presents the effectiveness of BDS and the proposed strategies through the simulation evaluation with the $PN^2$ model. Section 5 summarizes our key points and gives future work.

## 2. Related Work

### 2.1. Botnet and Mitigation Methods

Bots and botnets are not new technologies. A bot is a program that performs predefined tasks according to commands sent through the Internet. Some bots are good such as chat bots and trader bots, while others are bad such as spam bots and DDoS bots. A typical example of DDoS bots is Mirai. Mirai is a kind of worm that spreads copies of itself to IoT devices. Mirai infects IoT devices and turns them into bots. The network of bots (botnet) is used by attackers as a springboard for DDoS attacks [9].

Ogu et al. [10] surveyed the current status of the botnet threat. The survey covers the typology of botnets and their owners, the structure and life cycle of botnets, botnet attack modes and control architectures, existing countermeasure solutions and limitations, as well as the prospects of a botnet threat.

Bezerra et al. [11] proposed a host-based approach to detect IoT botnets, named IoTDS (Internet of Things Detection System). IoTDS monitors a device and collects its CPU use and temperature, memory consumption, and the number of processes. If the device detects any anomaly from the data, an alert of botnet detection is sent to the central server.

Manso et al. [12] designed and implemented a Software-Defined Intrusion Detection System. The system includes an IDS that automatically detects several DDoS attacks. Once the IDS detects an attack, it notifies a Software-Defined Networking controller to control networking devices. The system can timely detect a botnet exploitation, mitigate malicious traffic, and protect normal traffic. However, the above-mentioned systems do nothing for the detected botnets.

### 2.2. White-Hat Worm

A white-hat worm is generally defined as a worm created for well-intentioned purposes. One of well-known white-hat worms is Hajime [13]. Like Mirai, Hajime infects IoT devices and turns them into bots. However, there is neither code nor capability for DDoS attacks in Hajime. Moreover, Hajime actually protects the devices against Mirai by blocking the ports that Mirai

uses to infect. However, Hajime continues to stay at the infected device even after completing the defense against Mirai. Therefore, Hajime may be said to be gray-hat.

Molesky et al. [14] have proposed a new perspective in the use of white-hat worm technology. The point is that not only the manufacturers but also governments can use the technology to fix critical vulnerabilities of IoT devices. A white-hat worm infects the vulnerable device and applies the appropriate patch. Molesky states in Ref. [14] that legally, this could be enacted by including explicit terms within the Terms and Conditions agreement at the time of purchase and creates a contract with the consumer allowing for these actions to occur legally and without liability to the company. This suggests white-hat worm technology is applicable in practice. However, the role of the white-hat worm is restricted to fix vulnerabilities, not to exterminate malicious botnet.

Yamaguchi [7] defined a worm that drives out the Mirai botnet and deletes itself as a white-hat worm. It is characterized by two attributes: secondary infection possibility and lifespan. Secondary infection possibility is a probability that the worm can reinfect the devices infected by Mirai. This enables the worm to regain the device from Mirai. Lifespan is a temporal constraint that the worm can exist at a device. It forces the worm to destruct itself and to avoid staying on the device recovered. These characteristics make the white-hat worm radically different from Hajime.

### 2.3. PN² Model

Yamaguchi regarded the battle between Mirai and the white-hat worm as a multi-agent system and expressed it with agent-oriented Petri nets, called $PN^2$. For the detail of Petri nets and $PN^2$, refer to Refs. [8,9]. A $PN^2$ is intuitively a Petri net (known as *environment net*) whose tokens are again Petri nets (known as *agent nets*). Figure 1 shows a $PN^2$ model representing the battle. Figure 1a–e respectively show the agent nets representing devices `device1`, `device3`, Mirai, device `device2` and the white-hat worm. Figure 1f shows the environment net representing an IoT system composed of three network nodes. Each place drawn as ○ represents a network node. Each token drawn as ○ represents an agent such as Mirai, the white-hat worm, and an IoT device, which corresponds to one of the agent nets. Each node has one device. Device `device1` at place P1 is infected by Mirai. Device `device2` at place P2 is normal. Device `device3` at place P3 is infected by the white-hat worm. Each transition drawn as □ represents an action of one agent or an interaction among two or more agents. Whether an action can occur or not is decided based on the state of related agents. Red transitions of the environment net shown in Figure 1f show that they can occur. Let us induce the action represented by transition T214. This action means that the white-hat worm at P3 produces a copy of itself into P2, and the copy infects `device2`. Figure 2 shows the state after this action occurs. Please note that the action of transition T113 became disabled in order to occur. This means that the worm at P2 protects `device2` from Mirai at P1.

Yamaguchi performed simulation evaluation with a tool known as *PN2Simulator* [15]. In Ref. [7], he revealed the following properties between a white-hat worm's capability (lifespan $\ell$, secondary infection possibility $\rho$) and its effect: (i) if $\ell$ is short, the worm becomes extinct in course of time; (ii) if $\rho$ is high, the worm is effective regardless of $\ell$; and (iii) if $\ell$ is long, the worm is effective even if $\rho$ is low. However, there is no discussion about how to systematically launch the white-hat worm in response to Mirai's infection situation.

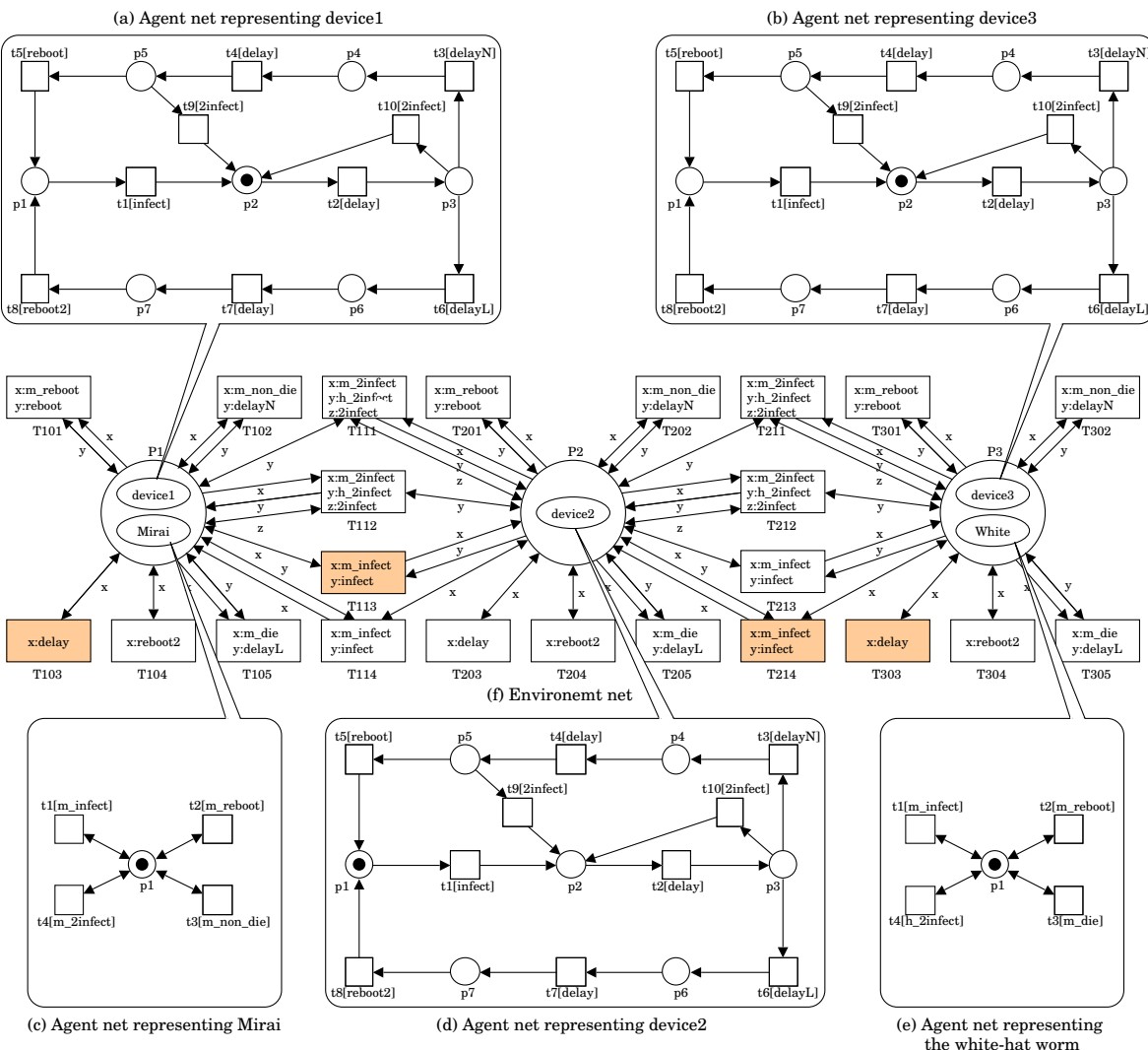

**Figure 1.** A PN$^2$ model representing a battle between Mirai and the white-hat worm.

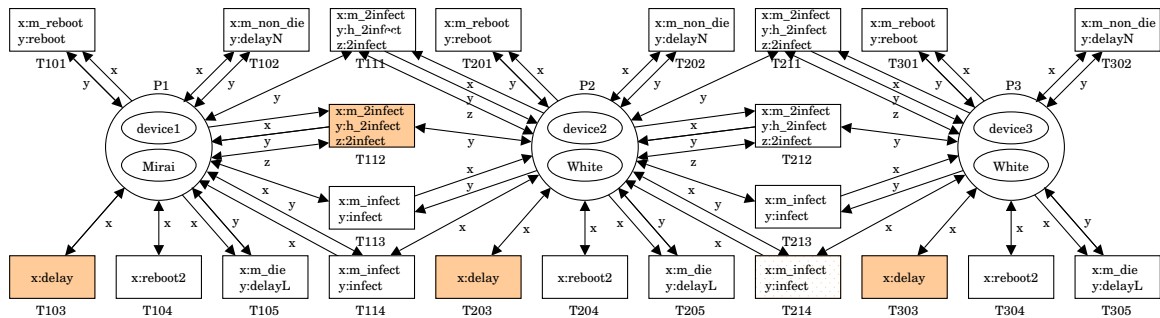

**Figure 2.** The state after the white-hat worm infected `device2`. This disables Mirai from infecting `device2`.

## 3. Botnet Defense System

### 3.1. Concept and Design

We propose a cyber-security system, named *Botnet Defense System* (BDS), that defends IoT systems against malicious botnets. Attackers use botnets to attack IoT systems. Imitating this, defenders also might be able to use botnets to defend the systems. This so-called "Fight fire with fire" is the concept of BDS. BDS realizes this concept and enables defenders to fight malicious botnets with white-hat botnet.

The distinct feature of BDS is to make use of the white-hat worm and its botnet. This can innovatively increase the defense ability because the white-hat worm and its botnet can protect devices from malicious botnet and drive it out instead of humans having to do that. However, since they are autonomous agents, we need to manage them to produce appropriate effect. BDS plans a strategy in response to malicious botnet's type and infection situation and carries out the management according to the strategy.

We adopt component-based architecture to design BDS. This enables us to research and develop the functionalities using components as a unit and further to realize a required BDS quickly and flexibly by combining components. A BDS consists of four components: *monitor*, *strategy planner*, *worm launcher*, and *command and control (C&C) server*. The BDS operates according to the following procedure (See Figure 3):

1°   The *monitor* component watches over a specified IoT system. This activity itself may be done through white-hat worm. If detecting a malicious botnet, it investigates and reports the information such as the botnet type and its infection situation.
2°   The *strategy planner* component makes a strategy against the malicious botnet based on the information reported by the monitor component.
3°   The *worm launcher* component sends white-hat worms into the IoT system based on the strategy and constructs a white-hat botnet.
4°   The *C&C server* component commands and controls the white-hat botnet to drive out the malicious botnet.

Please note that this procedure is executed repeatedly.

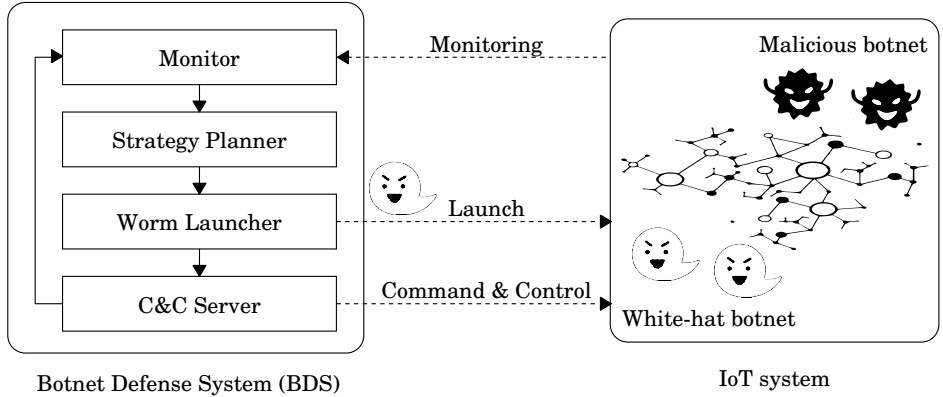

**Figure 3.** System configuration of BDS.

### 3.2. Strategies

The BDS uses a white-hat worm to build its botnet and then uses it to drive out the malicious botnet. The result would vary depending on the strategy adopted by the BDS. Therefore, strategy studies are essential to produce intended results. Strategies can be roughly divided into two categories: *launch strategies* and *C&C strategies*. Launch strategies specify how to launch a white-hat

worm to build its botnet. C&C strategies specify how to command and control the botnet to drive out malicious botnets. In this paper, we focus on the launch strategies because their result influences planning of the C&C strategies.

We first formalize the launch strategies. Once the monitor component detects a malicious botnet, it reports the information such as its type and infection situation. We assume that the malicious botnet is Mirai and its infection situation is grasped by its infection rate $R_{Mirai}$. $R_{Mirai}$ is given as

$$R_{Mirai} = \frac{\#_{Mirai}}{\#_{nodes}} \tag{1}$$

where $\#_{nodes}$ is the number of network nodes, $\#_{Mirai}$ is the number of nodes infected by Mirai, i.e., the number of Mirai bots. The strategy planner component makes a launch strategy based on the value of $R_{Mirai}$, the specification of the targeted IoT system, and the capability of an available the white-hat worm. An IoT system's specification is given as $\#_{nodes}$, the network topology $N_{topology} \in \{\text{Grid}, \text{Tree}, \cdots\}$, and the network density $N_{density}$. $N_{density}$ is given as

$$N_{density} = \frac{\#_{AC}}{\#_{PC}} = \frac{2 \cdot \#_{AC}}{\#_{nodes}(\#_{nodes} - 1)} \tag{2}$$

where $\#_{AC}$ is the number of actual connections and $\#_{PC}$ is the number of potential connections. A white-hat worm's capability is given as the worm's lifespan $\ell$ and secondary infection possibility $\rho$.

We give a formal definition of a launch strategy as follows.

**Definition 1** (Launch Strategy)**.** *Let $R_{Mirai}$ be a Mirai's infection rate, $(\#_{nodes}, N_{topology}, N_{density})$ be an IoT system's specification, and $(\ell, \rho)$ be a white-hat worm's capability. Let $\#_{White}$ be the number of the white-hat worm to launch. A launch strategy $\mathcal{L}$ is a mapping such that*

$$\mathcal{L} : (R_{Mirai}, (\#_{nodes}, N_{topology}, N_{density}), (\ell, \rho)) \mapsto \#_{White}. \tag{3}$$

### 3.2.1. All-Out Launch Strategy

In the beginning, we give a launch strategy called *All-Out* as a baseline in strategy studies. This is to launch as many white-hat worms as possible. The formal definition is given as follows.

**Strategy 1** (All-Out Launch Strategy $\mathcal{L}_{All-Out}$)**.** *For a Mirai's infection rate $R_{Mirai}$, an IoT system's specification $(\#_{nodes}, N_{topology}, N_{density})$ and a white-hat worm's capability $(\ell, \rho)$, the All-Out launch strategy $\mathcal{L}_{All-Out}$ is a mapping such that*

$$
\begin{aligned}
\mathcal{L}_{\text{All-Out}} : (R_{Mirai}, (\#_{nodes}, N_{topology}, N_{density}), (\ell, \rho)) &\mapsto \#_{nodes} - R_{Mirai}\#_{nodes} \\
&= \#_{nodes} - \#_{Mirai}
\end{aligned} \tag{4}
$$

This launch strategy $\mathcal{L}_{All-Out}$ is to dispatch the white-hat worm to all the nodes other than Mirai bots. Figure 4 shows an application example of $\mathcal{L}_{All-Out}$. This IoT system has a grid-topology network composed of $5 \times 5 = 25$ nodes, where the network density is 13.3% ($=(2 \times 40)/(25 \times 24)$), i.e., its specification $(\#_{nodes}, N_{topology}, N_{density}) = (25, \text{Grid}, 13.3\%)$. Figure 4a shows the state when the BDS detected a Mirai botnet. There are six Mirai bots, i.e., $\#_{Mirai} = 6$. According to $\mathcal{L}_{All-Out}$, the number of the white-hat worm to launch is calculated as $\#_{White} = \#_{nodes} - \#_{Mirai} = 19$. The BDS dispatches the worm to 19 non-bot nodes as shown in Figure 4b.

Obviously, the number ($\#_{nodes} - \#_{Mirai}$) of Equation (4) is the upper limit to launch. Therefore, when the BDS adopts this strategy, we can expect the white-hat worm and its botnet to expose the maximum effect against Mirai botnet. However, $\mathcal{L}_{All-Out}$ forces the BDS to launch the upper number of the white-hat worm regardless of the Mirai's infection situation, the IoT system's specification, and the worm's capability. Extra white-hat worms would place an unnecessary load on the IoT system.

A too-large white-hat botnet may make control difficult because all the network nodes may be not always observable and controllable.

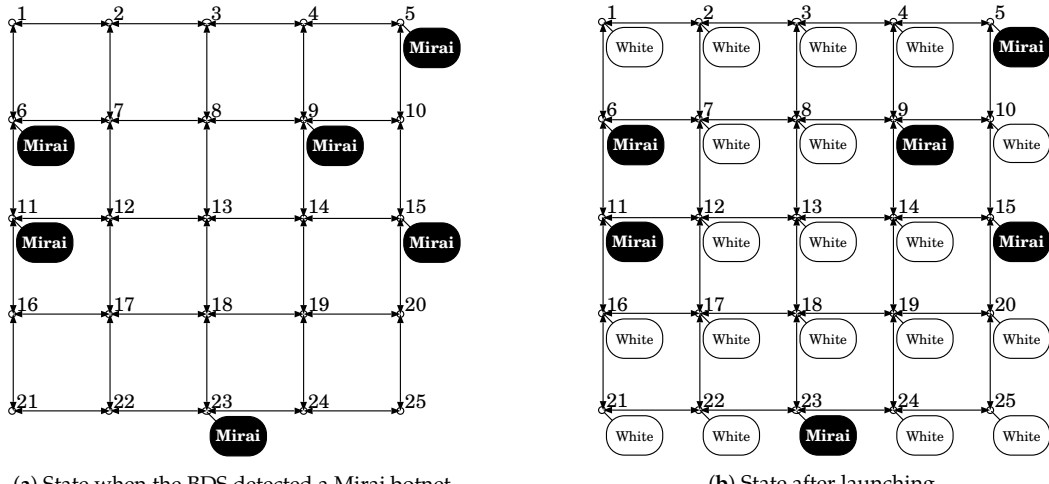

(**a**) State when the BDS detected a Mirai botnet.　　　　　　　(**b**) State after launching.

**Figure 4.** An application example of the All-Out launch strategy $\mathcal{L}_{All-Out}$. (**a**) State when the BDS detected a Mirai botnet, (**b**) State after launching.

### 3.2.2. Few-Elite Launch Strategy

Let us take the white-hat worm's capability (lifespan $\ell$, secondary infection possibility $\rho$) into consideration. As mentioned in Section 2.3, there are two conditions such that the worm becomes effective. One is $\rho$ is high. The other is $\ell$ is long. We propose a launch strategy called *Few-Elite* by introducing these conditions.

**Strategy 2** (Few-Elite Launch Strategy $\mathcal{L}_{Few\text{-}Elite}$)**.** *For a Mirai's infection rate $R_{Mirai}$, an IoT system's specification $(\#_{nodes}, N_{topology}, N_{density})$ and a white-hat worm's capability $(\ell, \rho)$, the Few-Elite strategy $\mathcal{L}_{Few\text{-}Elite}$ is a mapping such that*

$$\mathcal{L}_{Few\text{-}Elite} : (R_{Mirai}, (\#_{nodes}, N_{topology}, N_{density}), (\ell, \rho)) \;\mapsto\; \begin{cases} \#_{elite} & \text{if } \rho + \alpha\ell > \theta \\ \#_{nodes} - R_{Mirai}\#_{nodes} & \text{otherwise} \end{cases} \tag{5}$$

*where $\#_{elite}$ is the number of the worm to launch when its capability is sufficient, $\alpha$ is a weight coefficient and $\theta$ is a threshold.*

This launch strategy $\mathcal{L}_{Few\text{-}Elite}$ is to launch only a limited number $\#_{elite}$ if the worm has a high secondary infection possibility or a long lifespan. Otherwise, it launches as many worms as possible in the same as $\mathcal{L}_{All-Out}$. Figure 5 shows an application example of $\mathcal{L}_{Few\text{-}Elite}$. The IoT system used in this example and its infection situation are the same as that of Figure 4a. Assume that $\#_{elite} = 5$, $\alpha = 20$ and $\theta = 120$. Let an available worm's capability be $(\rho = 75\%, \ell = 3 \text{ steps})$. According to $\mathcal{L}_{Few\text{-}Elite}$, the number of the white-hat worm to launch is calculated as $\#_{White} = \#_{elite} = 5$ because $\rho + \alpha\ell = 75 + 20 \times 3 > \theta = 120$ holds. The BDS launches only five worms as shown in Figure 5a. As another case, assume that the worm's capability is $(\rho = 25\%, \ell = 3 \text{ steps})$. According to $\mathcal{L}_{Few\text{-}Elite}$, $\#_{White}$ is calculated as $\#_{nodes} - R_{Mirai}\#_{nodes} = \#_{nodes} - \#_{Mirai} = 19$ because $\rho + \alpha\ell = 25 + 20 \times 3 \ngtr \theta = 120$ holds. The BDS launches the upper number in the same as $\mathcal{L}_{All-Out}$ as shown in Figure 5b.

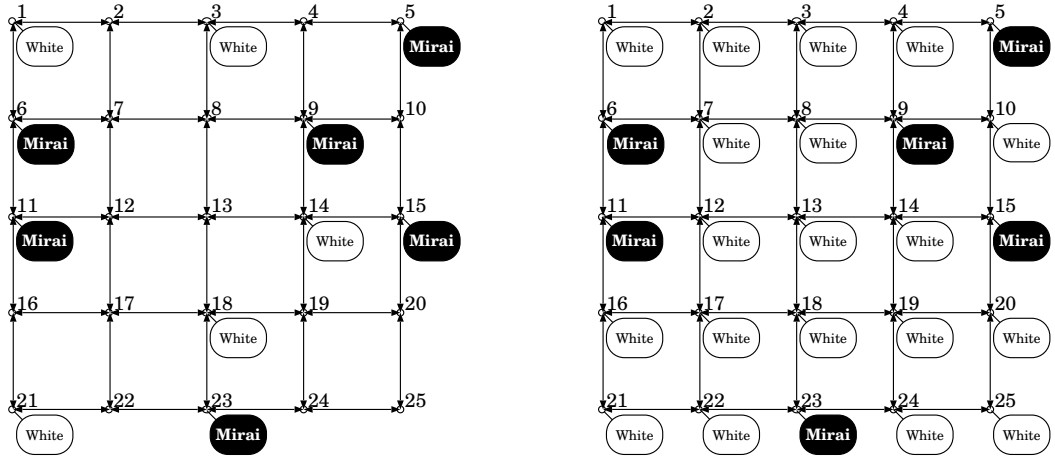

(**a**) State after launching a limited number when the worm's capability is sufficient.

(**b**) State after launching the upper number when the worm's capability is insufficient.

**Figure 5.** An application example of the Few-Elite launch strategy $\mathcal{L}_{Few-Elite}$, (**a**) State after launching a limited number when the worm's capability is sufficient, (**b**) State after launching the upper number when the worm's capability is insufficient.

### 3.2.3. Environment-Adaptive Strategy

Mirai and the white-hat worm use the network of an IoT system to spread themselves. Therefore, their spread is influenced by the network. We should introduce the influence to a strategy. The IoT system's specification is given as (Number of network nodes $\#_{nodes}$, Network topology $N_{topology}$, Network density $N_{density}$). We focus on $N_{density}$. This is because some networks have a different topology but the same density, and Ref. [7] indicated that $\#_{nodes}$ is not important than the worm's capability. We propose a launch strategy called *Environment-Adaptive* to compensate for the influence from the network.

**Strategy 3** (Environment-Adaptive Launch Strategy $\mathcal{L}_{Env-Adaptive}$). *For a Mirai's infection rate $R_{Mirai}$, an IoT system's specification $(\#_{nodes}, N_{topology}, N_{density})$ and a white-hat worm's capability $(\ell, \rho)$, the Environment-Adaptive strategy $\mathcal{L}_{Env-Adaptive}$ is a mapping such that*

$$\mathcal{L}_{Env-Adaptive} : (R_{Mirai}, (\#_{nodes}, N_{topology}, N_{density}), (\ell, \rho))$$

$$\mapsto \begin{cases} \#_{elite} & \text{if } \rho + \alpha\ell > \theta \text{ and } N_{density} > \beta\dfrac{2}{\#_{nodes}} \\ \#_{nodes} - R_{Mirai}\#_{nodes} & \text{otherwise} \end{cases} \quad (6)$$

*where $\frac{2}{\#_{nodes}}$ is the minimum density of connected networks composed of $\#_{nodes}$ nodes and $\beta$ is a weight coefficient.*

This launch strategy $\mathcal{L}_{Env-Adaptive}$ is to launch $\#_{elite}$ worms if the worm's capability is sufficient and only if the network density is not low. Otherwise, the upper number is launched. Figure 6 shows an application example of $\mathcal{L}_{Env-Adaptive}$. Let us consider two IoT systems. They have the same number of nodes ($\#_{nodes} = 25$) but have different network topology. One has the grid topology as shown in Figure 6a. The other has the tree topology as shown in Figure 6b. Let $\alpha = 20$ and $\theta = 120$ as well as the previous example. Assume that an available worm's capability is ($\rho = 75\%, \ell = 3$ steps). This worm has a sufficient capability because $\rho + \alpha\ell = 75 + 20 \times 3 > \theta = 120$ holds. Assume that $\#_{elite} = 5$ and $\beta = 1.2$. The threshold of network density is calculated as $\beta\frac{2}{\#_{nodes}} = 9.6\%$. First, let us consider the case of the grid topology. Since $\#_{AC} = 2\sqrt{\#_{nodes}}(\sqrt{\#_{nodes}} - 1) = 40$, $N_{density}$ is calculated as $\frac{2 \times 40}{25 \times 24} = 13.3\%$ by Equation (2). According to $\mathcal{L}_{Env-Adaptive}$, $\#_{White}$ is calculated as $\#_{elite} = 5$ because the worm's capability is sufficient and $N_{density} = 13.3\% > \beta\frac{2}{\#_{nodes}} = 9.6\%$. The BDS launches only five worms as

shown in Figure 6a. Next, let us consider the case of the tree topology. Since $\#_{AC} = \#_{nodes} - 1 = 24$, $N_{density}$ is calculated as $\frac{2 \times 24}{25 \times 24} = 8.0\%$ by Equation (2). According to $\mathcal{L}_{Env\text{-}Adaptive}$, $\#_{White}$ is calculated as the upper number because $N_{density} = 8.0\% \not\gtrsim \beta \frac{2}{\#_{nodes}} = 9.6\%$. Even though the worm has sufficient capability, the BDS launches the upper number to compensate the influence from the network as shown in Figure 6b.

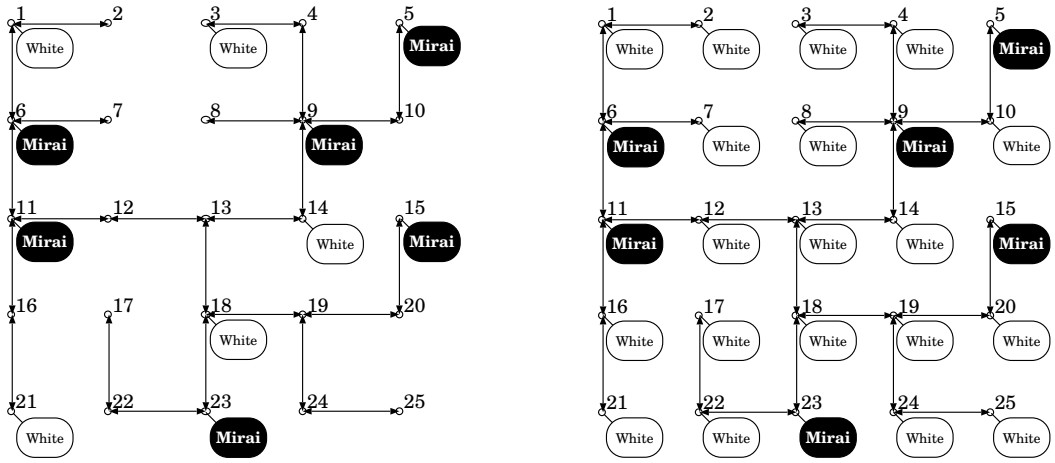

(**a**) State after launching a limited number when the network has the grid topology.  (**b**) State after launching the upper number when the network has the tree topology.

**Figure 6.** An application example of the Environment-Adaptive launch strategy $\mathcal{L}_{Env\text{-}Adaptive}$.

## 4. Simulation Evaluation

We evaluated BDS and the proposed launch strategies through the simulation of the PN$^2$ model representing the battle between Mirai botnet and the white-hat botnet.

### 4.1. Simulation

We performed a simulation experiment to evaluate BDS and the strategies with PN2Simulator [15]. The PN$^2$ model described in Section 2.3 can represent the dynamic behavior under various conditions. Figure 7 shows how to translate an IoT network into a PN$^2$ model. This network's specification is $(\#_{nodes}, N_{topology}, N_{density}) = (25, \text{Tree}, 8.0\%)$. Let us examine the 9th node. This node connects to four nodes: the 4th, 8th, 10th and 14th nodes. Each colored connection corresponds to the part of the PN$^2$ with the same color. Next, let us examine the 19th node. This node does not connect to the 14th node. Therefore, this would be translated into the model without the part colored in red.

We used $R_{Mirai}$ as the index to evaluate BDS and its strategies. The value of $R_{Mirai}$ varies with the progress of simulation. Therefore, we write it as a function $R_{Mirai}(t)$ of step number $t$, and set $t = 0$ when a BDS detects a Mirai botnet and launches the white-hat worms. For example, $R_{Mirai}(0)$ denotes Mirai's infection rate when the BDS detects the Mirai botnet. $R_{Mirai}(1000)$ denotes Mirai's infection rate after 1000 steps. In the same way, we define $\#_{Mirai}(t)$ and $\#_{White}(t)$, where $\#_{White}$ is the number of nodes infected by the white-hat worm, i.e., the number of the white-hat bots.

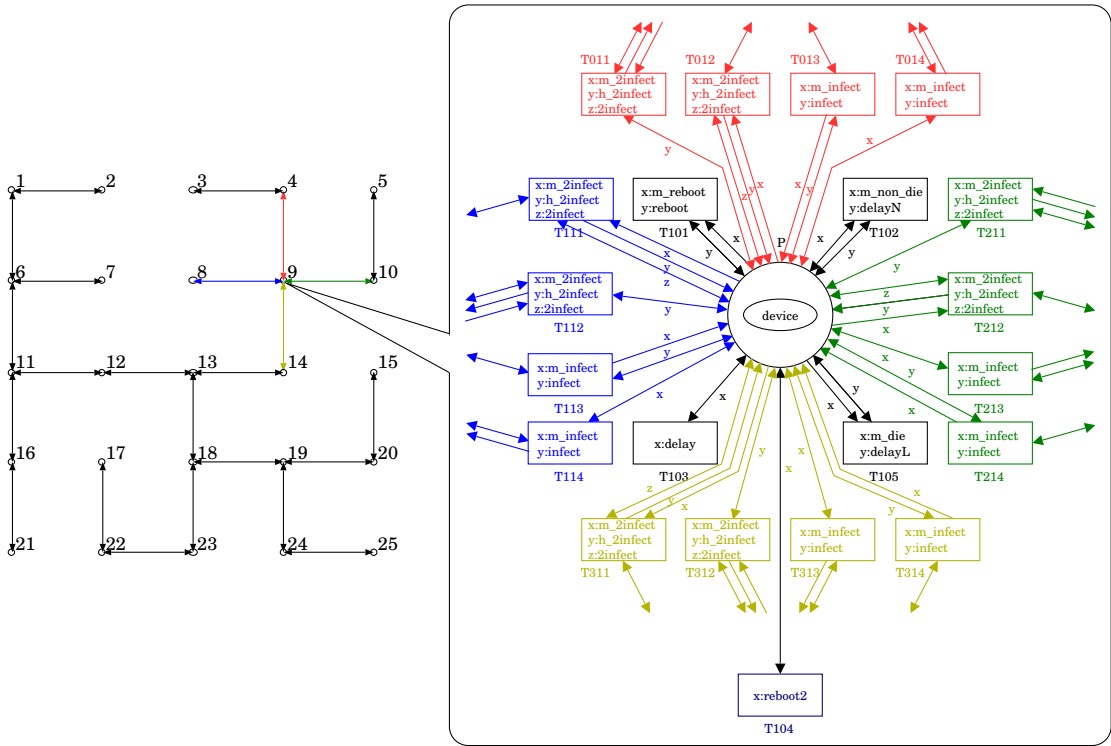

**Figure 7.** Illustration of translating a network into a PN$^2$ model.

We measured $R_{Mirai}(1000)$ by changing the following parameters.

- $\#_{Mirai}(0) = 6$ or $12$, i.e., $R_{Mirai}(0) = 24.0\%$ or $48.0\%$
- $\#_{White}(0) = 1, 3, 5, 7, 13$ (or $19$ if $\#_{Mirai}(0) = 6$)
- The distribution of Mirai and white-hat bots at step 0 were decided at random
- The IoT system's specification:
  - Network size $\#_{nodes} = 25$
  - Network topology $N_{topology} = $ Grid or Tree, i.e., Network density $N_{density} = 13.3\%$ or $8.0\%$
- The white-hat worm's capability:
  - Lifespan $\ell = 1, 3,$ or $5$ steps, where the delay time $\delta$ until rebooting = 11 steps
  - Secondary infection possibility $\rho = 0, 25, 50, 75,$ or $100\%$
- $\mathcal{L}_{Few\text{-}Elite}$:

$$(R_{Mirai}, (\#_{nodes}, N_{topology}, N_{density}), (\ell, \rho)) \mapsto \begin{cases} 5 & \text{if } \rho + 20\ell > 120 \\ \#_{nodes} - R_{Mirai}\#_{nodes} & \text{otherwise} \end{cases} \quad (7)$$

- $\mathcal{L}_{Env\text{-}Adaptive}$:

$$(R_{Mirai}, (\#_{nodes}, N_{topology}, N_{density}), (\ell, \rho)) \mapsto \begin{cases} 5 & \begin{aligned} &\text{if } \rho + 20\ell > 120 \text{ and} \\ &N_{density} > 1.2\frac{2}{\#_{nodes}} \end{aligned} \\ \#_{nodes} - R_{Mirai}\#_{nodes} & \text{otherwise} \end{cases} \quad (8)$$

The simulation result is shown in Tables 1–4. The first two are the cases of the grid topology and respectively show the result when $\#_{Mirai}(0) = 6$ or $12$, i.e., $R_{Mirai}(0) = 24.0\%$ or $48.0\%$. Similarly, the last two are that of the tree topology. Each table consists of three sub tables. The tables (a), (b) and (c) respectively show when $\ell = 1, 3,$ or $5$. Each cell shows the mean of $R_{Mirai}(1000)$ for 10,000 trials.

**Table 1.** Simulation result of applying BDS to the IoT system infected by Mirai botnet, where $\#_{Mirai}(0) = 6$, i.e., $R_{Mirai}(0) = 24.0\%$, $(\#_{nodes}, N_{topology}, N_{density}) = (25, \text{Grid}, 13.3\%)$. Yellow highlight: refer from the main text.

| Secondary Infection Possibility $\rho$ | Without BDS $\#_{White}(0)$ | | | | | | With BDS Launch Strategy | | |
|---|---|---|---|---|---|---|---|---|---|
| | 1 | 3 | 5 | 7 | 13 | 19 | $\mathcal{L}_{All\text{-}Out}$ | $\mathcal{L}_{Few\text{-}Elite}$ | $\mathcal{L}_{Env\text{-}Adaptive}$ |
| (a) $R_{Mirai}(1000)$ with $\ell = 1$ | | | | | | | | | |
| 0% | 97.0% | 97.0% | 96.9% | 97.0% | 96.5% | 96.9% | 96.9% | 96.9% | 96.9% |
| 25% | 96.9% | 96.9% | 96.7% | 96.5% | 95.5% | 94.7% | 94.7% | 94.7% | 94.7% |
| 50% | 94.0% | 88.9% | 82.9% | 77.2% | 66.5% | 48.1% | 48.1% | 48.1% | 48.1% |
| 75% | 70.0% | 61.5% | 53.9% | 49.5% | 41.8% | 27.7% | 27.7% | 27.7% | 27.7% |
| 100% | 37.4% | 26.1% | 21.9% | 17.6% | 13.8% | 8.6% | 8.6% | 8.6% | 8.6% |
| (b) $R_{Mirai}(1000)$ with $\ell = 3$ | | | | | | | | | |
| 0% | 97.0% | 96.9% | 96.9% | 97.0% | 96.5% | 96.8% | 96.8% | 96.8% | 96.8% |
| 25% | 84.1% | 78.1% | 75.6% | 74.1% | 71.4% | 68.0% | 68.0% | 68.0% | 68.0% |
| 50% | 32.6% | 29.7% | 27.2% | 25.2% | 24.0% | 15.7% | 15.7% | 15.7% | 15.7% |
| 75% | 7.5% | 6.6% | 6.2% | 5.9% | 6.7% | 4.0% | 4.0% | 6.2% | 6.2% |
| 100% | 1.3% | 0.6% | 0.4% | 0.4% | 0.1% | 0.2% | 0.2% | 0.4% | 0.4% |
| (c) $R_{Mirai}(1000)$ with $\ell = 5$ | | | | | | | | | |
| 0% | 96.8% | 96.5% | 96.0% | 95.8% | 94.6% | 93.4% | 93.4% | 93.4% | 93.4% |
| 25% | 22.0% | 11.4% | 10.1% | 9.7% | 7.9% | 8.9% | 8.9% | 10.1% | 10.1% |
| 50% | 1.5% | 0.9% | 1.0% | 1.0% | 0.8% | 0.7% | 0.7% | 1.0% | 1.0% |
| 75% | 0.2% | 0.1% | 0.2% | 0.2% | 0.1% | 0.1% | 0.1% | 0.2% | 0.2% |
| 100% | 0.0% | 0.0% | 0.0% | 0.0% | 0.0% | 0.0% | 0.0% | 0.0% | 0.0% |

**Table 2.** Simulation result of applying BDS to the IoT system infected by Mirai botnet, where $\#_{Mirai}(0) = 12$, i.e., $R_{Mirai}(0) = 48.0\%$, $(\#_{nodes}, N_{topology}, N_{density}) = (25, \text{Grid}, 13.3\%)$. Yellow highlight: refer from the main text.

| Secondary Infection Possibility $\rho$ | Without BDS $\#_{White}(0)$ | | | | | With BDS Launch Strategy | | |
|---|---|---|---|---|---|---|---|---|
| | 1 | 3 | 5 | 7 | 13 | $\mathcal{L}_{All\text{-}Out}$ | $\mathcal{L}_{Few\text{-}Elite}$ | $\mathcal{L}_{Env\text{-}Adaptive}$ |
| (a) $R_{Mirai}(1000)$ with $\ell = 1$ | | | | | | | | |
| 0% | 96.9% | 97.0% | 97.0% | 97.0% | 97.0% | 97.0% | 97.0% | 97.0% |
| 25% | 96.9% | 96.9% | 96.9% | 96.8% | 96.8% | 96.8% | 96.8% | 96.8% |
| 50% | 95.0% | 92.4% | 90.3% | 87.0% | 76.9% | 76.9% | 76.9% | 76.9% |
| 75% | 72.8% | 67.1% | 63.6% | 59.8% | 49.4% | 49.4% | 49.4% | 49.4% |
| 100% | 38.8% | 29.5% | 25.8% | 23.0% | 16.3% | 16.3% | 16.3% | 16.3% |
| (b) $R_{Mirai}(1000)$ with $\ell = 3$ | | | | | | | | |
| 0% | 97.0% | 97.0% | 97.0% | 96.9% | 96.9% | 96.9% | 96.9% | 96.9% |
| 25% | 86.4% | 79.9% | 79.6% | 78.4% | 77.4% | 77.4% | 77.4% | 77.4% |
| 50% | 35.0% | 32.1% | 32.3% | 31.0% | 27.0% | 27.0% | 27.0% | 27.0% |
| 75% | 7.9% | 8.1% | 8.4% | 8.1% | 7.4% | 7.4% | 8.4% | 8.4% |
| 100% | 1.1% | 0.7% | 0.6% | 0.7% | 0.5% | 0.5% | 0.6% | 0.6% |
| (c) $R_{Mirai}(1000)$ with $\ell = 5$ | | | | | | | | |
| 0% | 96.9% | 96.8% | 96.7% | 96.5% | 96.2% | 96.2% | 96.2% | 96.2% |
| 25% | 26.8% | 12.2% | 10.5% | 10.0% | 10.0% | 10.0% | 10.5% | 10.5% |
| 50% | 1.8% | 0.9% | 1.1% | 1.1% | 1.3% | 1.3% | 1.1% | 1.1% |
| 75% | 0.0% | 0.3% | 0.2% | 0.2% | 0.2% | 0.2% | 0.2% | 0.2% |
| 100% | 0.0% | 0.0% | 0.0% | 0.0% | 0.0% | 0.0% | 0.0% | 0.0% |

**Table 3.** Simulation result of applying BDS to the IoT system infected by Mirai botnet, where $\#_{Mirai}(0) = 6$, i.e., $R_{Mirai}(0) = 24.0\%$, $(\#_{nodes}, N_{topology}, N_{density}) = (25, \text{Tree}, 8.0\%)$. Yellow highlight: refer from the main text.

| Secondary Infection Possibility $\rho$ | Without BDS $\#_{White}(0)$ | | | | | | With BDS Launch Strategy | | |
|---|---|---|---|---|---|---|---|---|---|
| | 1 | 3 | 5 | 7 | 13 | 19 | $\mathcal{L}_{All\text{-}Out}$ | $\mathcal{L}_{Few\text{-}Elite}$ | $\mathcal{L}_{Env\text{-}Adaptive}$ |
| (a) $R_{Mirai}(1000)$ with $\ell = 1$ | | | | | | | | | |
| 0% | 94.5% | 94.4% | 94.6% | 94.4% | 94.4% | 93.9% | 93.9% | 93.9% | 93.9% |
| 25% | 94.4% | 94.5% | 94.4% | 94.3% | 94.0% | 92.4% | 92.4% | 92.4% | 92.4% |
| 50% | 94.5% | 94.1% | 92.9% | 91.1% | 81.2% | 67.9% | 67.9% | 67.9% | 67.9% |
| 75% | 94.4% | 92.5% | 89.0% | 85.4% | 70.6% | 57.2% | 57.2% | 57.2% | 57.2% |
| 100% | 93.1% | 87.9% | 80.2% | 72.8% | 52.8% | 38.1% | 38.1% | 38.1% | 38.1% |
| (b) $R_{Mirai}(1000)$ with $\ell = 3$ | | | | | | | | | |
| 0% | 94.5% | 94.5% | 94.5% | 94.4% | 94.3% | 93.9% | 93.9% | 93.9% | 93.9% |
| 25% | 94.3% | 94.0% | 93.2% | 92.3% | 87.8% | 82.9% | 82.9% | 82.9% | 82.9% |
| 50% | 92.6% | 87.4% | 80.9% | 74.4% | 59.4% | 45.8% | 45.8% | 45.8% | 45.8% |
| 75% | 81.2% | 66.0% | 54.8% | 48.0% | 33.6% | 24.1% | 24.1% | 54.8% | 24.1% |
| 100% | 58.9% | 40.3% | 31.1% | 24.7% | 13.7% | 7.7% | 7.7% | 31.1% | 7.7% |
| (c) $R_{Mirai}(1000)$ with $\ell = 5$ | | | | | | | | | |
| 0% | 94.2% | 93.8% | 93.6% | 93.4% | 92.3% | 91.6% | 91.6% | 91.6% | 91.6% |
| 25% | 85.7% | 76.9% | 69.4% | 64.6% | 54.3% | 47.2% | 47.2% | 69.4% | 47.2% |
| 50% | 57.7% | 40.2% | 31.7% | 26.9% | 19.1% | 14.4% | 14.4% | 31.7% | 14.4% |
| 75% | 27.4% | 13.9% | 10.3% | 8.1% | 5.9% | 3.9% | 3.9% | 10.3% | 3.9% |
| 100% | 9.5% | 4.6% | 2.9% | 2.5% | 1.2% | 0.6% | 0.6% | 2.9% | 0.6% |

**Table 4.** Simulation result of applying BDS to the IoT system infected by Mirai botnet, where $\#_{Mirai}(0) = 12$, i.e., $R_{Mirai}(0) = 48.0\%$, $(\#_{nodes}, N_{topology}, N_{density}) = (25, \text{Tree}, 8.0\%)$. Yellow highlight: refer from the main text.

| Secondary Infection Possibility $\rho$ | Without BDS $\#_{White}(0)$ | | | | | With BDS Launch Strategy | | |
|---|---|---|---|---|---|---|---|---|
| | 1 | 3 | 5 | 7 | 13 | $\mathcal{L}_{All\text{-}Out}$ | $\mathcal{L}_{Few\text{-}Elite}$ | $\mathcal{L}_{Env\text{-}Adaptive}$ |
| (a) $R_{Mirai}(1000)$ with $\ell = 1$ | | | | | | | | |
| 0% | 94.4% | 94.3% | 94.5% | 94.4% | 94.5% | 94.5% | 94.5% | 94.5% |
| 25% | 94.4% | 94.5% | 94.4% | 94.5% | 94.5% | 94.5% | 94.5% | 94.5% |
| 50% | 94.4% | 94.4% | 94.3% | 93.9% | 90.0% | 90.0% | 90.0% | 90.0% |
| 75% | 94.3% | 93.7% | 92.6% | 91.3% | 83.9% | 83.9% | 83.9% | 83.9% |
| 100% | 93.6% | 91.1% | 87.3% | 82.8% | 66.8% | 66.8% | 66.8% | 66.8% |
| (b) $R_{Mirai}(1000)$ with $\ell = 3$ | | | | | | | | |
| 0% | 94.4% | 94.4% | 94.5% | 94.4% | 94.5% | 94.5% | 94.5% | 94.5% |
| 25% | 94.3% | 94.2% | 94.2% | 93.8% | 92.0% | 92.0% | 92.0% | 92.0% |
| 50% | 93.1% | 90.4% | 87.5% | 83.7% | 72.5% | 72.5% | 72.5% | 72.5% |
| 75% | 82.5% | 71.2% | 64.0% | 59.4% | 45.7% | 45.7% | 64.0% | 45.7% |
| 100% | 61.8% | 45.7% | 38.8% | 33.4% | 20.2% | 20.2% | 38.8% | 20.2% |
| (c) $R_{Mirai}(1000)$ with $\ell = 5$ | | | | | | | | |
| 0% | 94.4% | 94.3% | 94.1% | 94.1% | 93.8% | 93.8% | 93.8% | 93.8% |
| 25% | 87.5% | 79.2% | 74.2% | 70.4% | 62.5% | 62.5% | 74.2% | 62.5% |
| 50% | 60.6% | 43.5% | 36.7% | 32.7% | 23.9% | 23.9% | 36.7% | 23.9% |
| 75% | 28.1% | 15.8% | 13.0% | 10.4% | 8.0% | 8.0% | 13.0% | 8.0% |
| 100% | 9.8% | 5.2% | 4.3% | 3.6% | 1.9% | 1.9% | 4.3% | 1.9% |

*4.2. Discussion*

First, let us discuss the case of the grid topology. Let us see Table 1. This shows the result when $\#_{Mirai}(0) = 6$, i.e., $R_{Mirai}(0) = 24.0\%$, and $(\#_{nodes}, N_{topology}, N_{density}) = (25, \text{Grid}, 13.3\%)$. $R_{Mirai}(1000)$ decreased with the increase of $\#_{White}(0)$. It also decreased with the increase of secondary infection possibility $\rho$. Comparing Table 1a–c and seeing the influence of lifespan $\ell$, we found that $R_{Mirai}(1000)$ decreased with the increase of $\ell$. Next, let us see Table 2. This shows the result when the initial number of Mirai bots is twice as much as Table 1. Each value became slightly bigger because the initial number of Mirai bots increased but the trend was the same.

Let us discuss the effect of the proposed strategies. $R_{Mirai}(1000)$ almost monotonously decreased with the increase of $\#_{White}(0)$. This result backs up $\mathcal{L}_{All\text{-}Out}$'s validity. However, white-hat bots are a so-called double-edged sword. They defend the IoT system against Mirai bots, but they waste the system's resources if they stay there even after exterminating all the Mirai bots. $\mathcal{L}_{Few\text{-}Elite}$ and $\mathcal{L}_{Env\text{-}Adaptive}$ can reduce $\#_{White}(0)$ than $\mathcal{L}_{All\text{-}Out}$. In Table 1, the yellow highlighted cells show the results when only 5 worms were launched instead of 19 ones. When the worm's lifespan is short (See Table 1a), i.e., $\ell = 1$, all the strategies launched 19 worms because the worm's capability was decided to be insufficient according to Equation (7). When $\ell = 3$ and $\rho \geq 75\%$ (See Table 1b), $\mathcal{L}_{Few\text{-}Elite}$ and $\mathcal{L}_{Env\text{-}Adaptive}$ launched only five worms. Those strategies reduced $\#_{White}(0)$ from 19 to 5 only after increasing $R_{Mirai}$ by at most 2.2%. When $\ell = 5$ and $\rho \geq 25\%$ (See Table 1c), $\mathcal{L}_{Few\text{-}Elite}$ and $\mathcal{L}_{Env\text{-}Adaptive}$ reduced $\#_{White}(0)$ from 19 to 5 almost without changing $R_{Mirai}$. Moreover, as shown in Table 2, even though the initial number of Mirai bots is twice, if a given worm has enough capability, $\mathcal{L}_{Few\text{-}Elite}$ and $\mathcal{L}_{Env\text{-}Adaptive}$ reduced $\#_{White}$ to five only with increasing $R_{Mirai}$ by at most 1.0%. From the above, we can say that $\mathcal{L}_{Few\text{-}Elite}$ and $\mathcal{L}_{Env\text{-}Adaptive}$ are effective for the IoT systems with the grid topology.

Next, let us discuss the case of the tree topology. Let us see Table 3. This shows the result when $\#_{Mirai}(0) = 6$, i.e., $R_{Mirai}(0) = 24.0\%$, and $(\#_{nodes}, N_{topology}, N_{density}) = (25, \text{Tree}, 8.0\%)$. $R_{Mirai}(1000)$ decreased with the increase of $\#_{White}(0)$, $\rho$ or $\ell$. However, in comparison to the result of Table 1, the reduction rate of $R_{Mirai}(1000)$ significantly reduced. This means that the network with low density restrains the white-hat bots' activities. Next, let us see Table 4. This shows the result when the initial number of Mirai bots is twice as much as Table 3. In the same way as the case of the grid topology, each value became slightly bigger, but the trend was the same.

Next, let us discuss the effect of the proposed strategies. In Tables 3 and 4, the yellow highlighted cells show the results when only 5 worms were launched. As with the case of the grid topology, $\mathcal{L}_{Few\text{-}Elite}$ reduced $\#_{White}(0)$ to five based on the worm's capability only. Meanwhile, $\mathcal{L}_{Env\text{-}Adaptive}$ did not reduce $\#_{White}(0)$ under that condition because the network density is low, i.e., $N_{density} = 8.0\% \ngtr 1.2\frac{2}{\#_{nodes}} = 1.2\frac{2}{25} = 9.6\%$. We should discuss the advantage and disadvantage. For instance, let us see the result when $ell = 3$ and $\rho = 75\%$ in Table 3b. $\mathcal{L}_{Few\text{-}Elite}$ reduced $\#_{White}(0)$ to five. However, $R_{Mirai}(1000)$ increased by 30.7%. So did the other highlighted cells. We think that five worms was too few to produce enough effect under the restriction caused by the network with low density. From the above discussion, we can say that $\mathcal{L}_{Env\text{-}Adaptive}$ is reasonable for the IoT systems with the tree topology.

## 5. Conclusions

In this paper, we proposed the concept, design and basic strategies of Botnet Defense System (BDS). Imitating "Fight fire with fire", we advocated the concept of BDS, "Fight botnet with botnet". When a BDS detects a malicious botnet, it plans a strategy against the botnet. Based on the planned strategy, the BDS sends a white-hat worm and builds its botnet on the IoT system. The BDS uses the white-hat botnet to exterminate the malicious botnet. We adopted component-based architecture and composed BDS from four components. Next, we studied a strategy to launch white-hat worms. We formalized a launch strategy and proposed three kinds of concrete strategies. The All-Out strategy is a baseline in strategy studies and sends white-hat worms to all the non-bot nodes. The Few-Elite

strategy can reduce white-hat worms according to its capability. The Environment-Adaptive strategy can adjust the number of white-hat worms based on not only the worm's capability but also the IoT system's specification. We modeled the battle between Mirai botnets and the white-hat botnets with agent-oriented Petri net $PN^2$ and evaluated BDS and the proposed strategies through the simulation of the $PN^2$ model. The simulation result shows that the Environment-Adaptive strategy is the best and reduced the number of needed white-hat worms to 38.5% almost without changing the extermination rate for Mirai bots.

In future work, we will propose C&C strategies and combine them with the proposed launch strategies.

**Funding:** This research was funded by JSPS KAKENHI Grant Number JP19K11965.

**Conflicts of Interest:** The author declares no conflict of interest.

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
