# Peer review of "Botnet Defense System: Concept, Design, and Basic Strategy†"

_information, doi:10.3390/info11110516_

Round 1
Reviewer 1 Report
The paper is well written, job well done.
Author Response
Thank you for your review carefully.
I revised the English expression.

Reviewer 2 Report
Interesting paper, which needs some modifications. Firstly, the conclusion is exact copy and paste of abstract. It must be rewritten. Secondly, the simulation evaluation is very difficult to understand and it is missing important conclusions (discussion). Please modify this section accordingly. Thirdly, please address the following comments:
1. almost without changing the effect (14, 303) – what effect?
2. Once detecting an attack, the IDS invokes a Software Defined Networking controller. (70) – SDN controller cannot be invoked – in every SDN it must run in order to control networking devices.
3. Red transitions show that they can occur. (107) – there are no colors in the figure.
4. Figures are included before they are referenced in text.
5. secondary infection possibility (254) - define this term
6. Check references format.
Finally, the paper need proof-reading as it contains a lot of grammatical errors and some sentences are hard to understand. For example:
1. In addition, IoT devices are vulnerable. (29)
2. avoid staying on the device recovered. (39)
3. A BDS (146, 171) – should be THE
4. strategy studies is (147) – are
5. have different network topology (212) – a
6. without the red part (234)
7. are twice (270) – is twice
8. with density low restrains (278) – low density?
Author Response
Thank you for your review carefully.
* Firstly, the conclusion is exact copy and paste of abstract. It must be rewritten.
--> I rewrote the conclusion.
Secondly, the simulation evaluation is very difficult to understand and it is missing important conclusions (discussion). Please modify this section accordingly.
--> I modified Section 4 (Simulation evaluation). I separated and made the discussion clear.
Thirdly, please address the following comments:
1. almost without changing the effect (14, 303) – what effect?
--> It means the reduction rate for Mirai bots. I clearly wrote it.
2. Once detecting an attack, the IDS invokes a Software Defined Networking controller. (70) – SDN controller cannot be invoked – in every SDN it must run in order to control networking devices.
--> You are right. I corrected as follows: Once the IDS detects an attack, it notifies a Software Defined Networking controller to control networking devices.
3. Red transitions show that they can occur. (107) – there are no colors in the figure.
--> Transitions highlighted in red are only of the environment net shown in Figure 1 (e). I clearly wrote that.
4. Figures are included before they are referenced in text.
--> I checked that.
5. secondary infection possibility (254) - define this term
--> I defined it in Line 89.
6. Check references format.
--> I revised it.
Finally, the paper need proof-reading as it contains a lot of grammatical errors and some sentences are hard to understand. For example:
According to your indication, I corrected all:
1. In addition, IoT devices are vulnerable. (29)
2. avoid staying on the device recovered. (39)
3. A BDS (146, 171) – should be THE
4. strategy studies is (147) – are
5. have different network topology (212) – a
6. without the red part (234)
7. are twice (270) – is twice
8. with density low restrains (278) – low density?
Thank you so much for your useful comments.

Reviewer 3 Report
The presented paper is well written and present a work clearly and in a suitable way.
The pape presents a concept, design and some strategies of BDS, which is a R&D hot topic. It's an interesting and solid work.
Author Response
Thank you for your review carefully.

Round 2
Reviewer 2 Report
Thank you for correcting my comments.